# CHIRon: A Generative Foundation Model for Structured Sequential Medical Data

## Abstract

Recent advances in large language models (LLMs) have shown that foundation models (FMs) can learn highly complex representations of sequences that can be used for downstream generative and discriminative tasks such as text generation and classification. While most FMs focus on text, recent work has shown FMs can be learnt for sequential medical data, e.g. ICD-10 diagnosis codes associated with specific patient visits. These FMs demonstrate improved performance on downstream discriminative disease classification tasks. In this paper, we introduce CHIRon, a decoder-only generative FM for sequential medical data. CHIRon utilizes causal masking during pre-training, enabling generative applications, and incorporates a number of architectural improvements and support for additional medical data types (diagnoses, procedures, medications, lab results, place of service, demographics). We introduce a new pre-training objective function that incorporates tasks for predicting place of service and patient's age at encounter in addition to the next medical code prediction task. To incorporate lab results into the model, we develop and evaluate several methods for embedding the continuous lab values. Furthermore, we introduce a causal visit-based masking approach for training CHIRon based on patient visits. We show empirically that CHIRon can be used to generate realistic sequential medical data and also outperforms state of the art FMs for sequential medical data on disease classification tasks.

## 1 Introduction

Foundation models (FMs) offer many improvements over traditional machine learning (ML) models, including better predictive performance, requiring less labeled data, and simplifying model deployment (Wornow et al., 2023). However, most prior work using FMs in the healthcare setting focuses on text such as clinical notes (Huang et al., 2019) or biomedical text (Lee et al., 2020), despite significant amounts of healthcare data such as administrative claims or electronic health records (EHRs) being stored in structured databases.

Several papers have developed FMs such as BERT (Devlin et al., 2019) using structured sequential medical data (Rasmy et al., 2021; Li et al., 2020), such as ICD-10 diagnosis codes associated with specific patient visits, and have shown promising improvements over traditional ML methods in downstream prediction tasks such as disease classification. These BERT-based FMs, however, cannot easily be used for generative purposes (Patel et al., 2023) – for example, generating synthetic visit sequences to enable privacy-preserving data sharing applications or augmenting existing patient data (Zhang et al., 2022). Given the recent success of generative FMs such as GPT-style models for text (Radford et al., a;b; Brown et al., 2020), we propose a novel generative FM for structured sequential patient data and investigate its performance on both generative and discriminative tasks.

In this work, we introduce CHIRon (Contextualized Healthcare Information RepresentatiON), a decoder-only generative FM trained on structured sequential medical data (rather than text). CHIRon includes a number of architectural improvements, support for additional data types, and a new objective function for pre-training that incorporates additional tasks beyond next code prediction. Unlike previous transformer-based models for sequential medical data that have focused specifically on diagnosis codes, we expand to include procedure codes, medications, lab results, and patient demographics for additional context. We also implement multiple methods for incorporating continuous lab results into our model. By adding extra task heads to the model for predicting the place-of-service

and age-at-encounter information, we enabled CHIRon to simultaneously generate these sequences alongside the medical code sequences. Furthermore, we experiment with a new visit-based masking approach (instead of the traditional causal masking approach) where only codes from the previous and current visits can be used for predicting the codes in the next visit. Fine-tuning CHIRon for disease onset and progression classification tasks shows it outperforms existing state-of-the-art discriminative FMs for sequential medical data. CHIRon also demonstrates strong generative capabilities, as evaluated using several quantitative metrics, proving generative FMs are powerful for generating and classifying sequential medical data.

**Summary of contributions:**

- We introduce CHIRon, a decoder-only generative FM trained on structured sequential medical data. In addition to diagnosis codes, we include procedure codes, medications, lab results, and patient demographics.

- We introduce a novel embedding for place-of-service information that adds useful context to each medical code.

- We develop methods for handling continuous lab results including using a shared decile embeddings as well as scaling the lab code embeddings using the continuous values.

- We incorporate additional task heads in the model architecture for predicting place-of-service and age-at-encounter information during pretraining. We empirically show that this new objective improves the performance of the model for sequential code generation.

- We propose a novel visit-based causal masking apprach that ensures only codes from the current or previous visits are used to predict codes in the next visit.

- Fine-tuning our FM shows improvements over state-of-the-art models on downstream disease onset and progression classification tasks.

- We demonstrate CHIRon's generative capabilities for creating realistic patient records. We are able to simultaneously generate important medical context such as place-of-service and age-at-encounter information to augment the medical code sequence. We evaluate the generative performance using metrics such as the BERTScore and the ROUGE score.

## 2 RELATED WORK

**Language model pretraining**   LLM pre-training has shown remarkable success in a variety of downstream tasks. These models efficiently use in-context information and eliminate the need for task-specific architectures. One of the most widely used models, BERT (Devlin et al., 2019), is built on the Transformer (Vaswani et al., 2017) architecture and uses bidirectional context for learning representations. The core training objective employed by BERT is masked language modeling which encourages the model to better understand word relationships. GPT-style models (Radford et al., a;b; Brown et al., 2020) similarly uses a Transformer architecture but emphasize auto-regressive generation, which is useful in synthetic data generation. It scales up to billions of parameters and can perform both conditional and unconditional text generation. Like BERT, it can also be adapted for different NLP tasks by fine-tuning. Recent works have incorporated BERT and GPT for NLP tasks using medical text such as Lee et al. (2020); Luo et al. (2022); Gu et al. (2021); Alsentzer et al. (2019); Huang et al. (2019); Yang et al. (2022).

**Representation learning frameworks in the clinical domain**   One successful model that leveraged the temporal dependencies in clinical events is RETAIN (Choi et al., 2016), which is an RNN-based model that uses a two-level neural attention mechanism for learning visit representations. Models such as BEHRT (Li et al., 2020) and G-BERT (Shang et al., 2019) have attempted to employ contextualized pre-trained embeddings in the clinical domain. The former developed a model for diagnosis code prediction in different time windows and the latter leveraged graph neural networks for medication code prediction using a single-visit-level dataset. One of the most relevant works to ours is Med-BERT (Rasmy et al., 2021). The authors train a BERT model to learn contextualized diagnosis code embeddings to use for downstream disease prediction tasks. They utilize visit and/or positional embeddings in addition to the codes embeddings in their architecture. While BERT models

are able to effectively learn contextualized code representations, they are not explicitly optimized for generation tasks like GPT-style transformer models.

CLMBR (Steinberg et al., 2021) proposed an auto-regressive Transformer-based (and also a GRU-based) foundation model for EHR data which is pre-trained to predict a patient's next day codes. The model is then used to generate feature representations for downstream tasks using a logistic regression head with the main purpose of comparing the in- and out-of-distribution performance to models trained on count-based representations. One important distinction between the CLMBR model and CHIRon is that the CLMBR model is applied to regularly-sampled data over a relatively short time duration (e.g., at the granularity of a day during inpatient/ICU hospital stays) of patient history, whereas CHIRon (and other methods such as Med-BERT) operate on irregularly-sampled data over much longer time durations (e.g., encounter dates distributed throughout many months/years).

Recently, Yang et al. (2023) introduced TransformEHR, a generative encoder-decoder model that predicts patients' next visit diagnosis codes. Similar to our work, TransformEHR incorporates temporal embeddings along with visit embeddings in their architecture. While the TransformEHR model uses only diagnosis codes, our framework integrates additional medical data types such as procedure codes and medications, continuous lab results, and contextual information such as place of service. We also include extra tasks in our pre-training objective for predicting the place of service and patient's age at encounter. In this paper, we adapt the framework of GPT-style decoder-only models and pre-train our model on structured health records (rather than clinical text).

## 3 METHODS

For our experiments we utilized a large healthcare institution's de-identified data[1] which contains structured administrative claims and clinical data such as medical and pharmacy claims, lab results, demographics, and enrollment records for 44 million patients. The protocol and supporting materials representing this work were prospectively submitted to the [REDACTED] for IRB review and were approved. We extracted demographics (age and sex), diagnosis codes, procedure codes, medications, and lab results, along with their corresponding encounter dates and place-of-service information, to build chronologically ordered lists of medical codes for each individual. See Appendix A for more details.

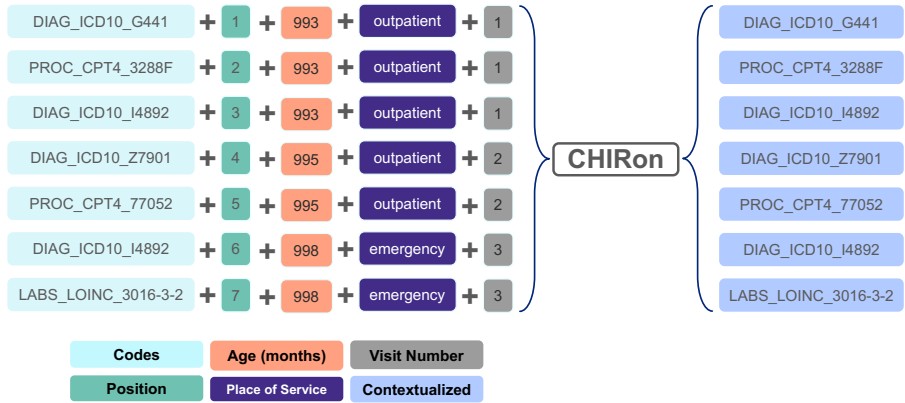

Figure 1: Diagram of embeddings used in CHIRon model.

**CHIRon pre-training** CHIRon is a GPT-style (Radford et al., a;b) model, where each medical code is represented as an individual token. We augment the GPT architecture with several additional embeddings to add healthcare-specific context to each code. In addition to the standard positional embeddings, we include visit embeddings (similar to Rasmy et al. (2021); Li et al. (2020)), age embeddings (similar to Li et al. (2020)), and place-of-service embeddings, a novel data type. Place of service specifies where the service (code) took place, including these locations: outpatient, inpatient,

---

[1]To comply with the double-blind submission policy we withhold the name of the institution. We will reveal it should the paper be accepted.

emergency, custodial, independent lab, home (or unknown). This information adds important context – e.g., a diagnosis code for chest pain should have a different representation if it occurred in a primary care office versus an emergency room setting. Each of these embeddings are element-wise added to the code embeddings before being used as input to the model. Figure 1 presents a comprehensive diagram of all the embeddings used in the model. We also prepend two tokens to every code sequence: one token indicating the sex of the patient and one token indicating the patient's age (binned into 5-year groups – e.g., 20-24, 25-29, etc.). The model was pre-trained using the causal language modeling objective as described in Radford et al. (b).

## 3.1 PREPROCESSING

### 3.1.1 TOKENIZATION:

We built a tokenizer similar to Rasmy et al. (2021), where each token corresponds to a unique medical code, using the Combined Dataset. Rather than use a vocabulary that included all 60,817 unique medical codes in the combined dataset, we selected codes that occurred in at least 1/1000 individuals, given that many codes are rare and only occur in a small subset of patients. This prevalence-based filtering left us with a tokenizer vocabulary of 7,922 unique medical codes. Any medical code that did not pass this prevalence-based threshold was not discarded but instead renamed as a "rare code" that was specific to the type of medical code (e.g., "DIAG_ICD10_RARE_CODE" or "PROC_CPT4_RARE_CODE").

### 3.1.2 LAB RESULTS:

Previous methods likely ignored lab results because they are noisy and need to be converted into discrete tokens. However, lab results contain specific and objective information about patient state, and can be used to more accurately phenotype patients for disease labeling. To incorporate the continuous lab results into our modeling framework, we explored 3 different tokenization methods:

**Tokenization per lab code per decile bin:**   In this approach, we select only LOINC codes with at least 1000 observations to remove rare codes. We then compute deciles for each LOINC code and drop LOINC codes where the max value for third decile is 0, to remove labs where a significant number of results were zero-filled by an upstream data management process as the result data was unavailable. We then fit an exponential function using the maximum values from first 9 deciles and use this function to compute the 10th decile. Finally we drop observations (tokens) which are more than $3\times$ the predicted 10th decile. Therefore, each lab result token used as input to our model denotes the decile of the lab result for that specific test. For example, for a "Hemoglobin A1c/Hemoglobin - total in Blood" lab result (LOINC 4548-4) that fell into the 7th decile of the population distribution, we would denote the lab result token as "LABS_LOINC_4548-4-7" where the decile is added as a suffix to the token ID. Similar to the deciles, we experimented with choosing different percentile ranges as bins for the lab tokens. Table 15 shows the bins corresponding to each percentile range based on the number of observed records for each lab. The purpose of this experiment was to create more meaningful ranges for each bin as extreme high/low lab results could potentially help pick up on certain conditions.

**Tokenization per lab code per bin with embedding scaling:**   Inspired by Golkar et al. (2023), we propose a method for handling continuous lab values by scaling the lab code embeddings. In this approach, in addition to the lab token (including the bin), we provide the model with the normalized lab values as an additional input vector. The continuous lab values are min-max normalized based on the low and high end values of the corresponding bin and then mapped to the interval [1,2]. Each lab token embedding is then scaled element-wise by theses normalized values and used as input to the pre-training task. In addition to the LM head in the model, we add a new head that predicts the continuous value for each lab token and calculates the mean squared error (MSE) loss. The combined loss is then optimized during the pre-training task.

**Tokenization per lab code and with shared decile embeddings:**   We developed another method to reduce the vocabulary size for tokenizing continuous values such as lab results using separate decile embeddings. In this approach, we use two separate sets of embeddings: each lab code is represented by a single token, and each individual decile is represented by a single token. The decile embeddings

are shared across all lab codes. The decile bins are calculated based on the splits computed for the pre-training set (similar to the per-lab-code per-decile bin tokenization scheme) and are given to the model as a separate input vector. The final embedding used by the model for each lab code is the sum of the lab code embedding and the decile bin embedding. Similar to the LM head, we add an additional head to the FM to predict the decile bin token whenever the next predicted code is a lab token. The model is optimized to minimize the sum of original LM loss and the cross-entropy loss for predicting the decile token. A visualization of all three methods is presented in Figure 8 in the appendix.

## 3.2 Model Development

CHIRon is a GPT-based model and we adopt a similar architecture and pre-training techniques as GPT-2 and build on top of them.

**Input Representation:** A patient record consists of a series of encounters (i.e. visits), each containing several medical code tokens including diagnosis, procedure, and medication codes as well as tokenized lab results as explained above. Let $\mathbf{x}_c = (c_{11}, \cdots, c_{1n_1}, \cdots, c_{K1}, \cdots, c_{Kn_K})$ be the code sequence for patient $X$ with $K$ total visits where $c_{ij}$ corresponds to the $j$-th code (token) that occurred in visit $i$, $i \in [K], j \in [n_i]$ where $n_i$ is the total number of codes for visit $i$. For each encounter, the information about its place of service (one of a total of 7 categories), its timestamp which represents age of the patient in months at the time of encounter, and the visit number in clinical history is available as well. Given the total code sequence has $N$ codes, let the following denote the context information sequences:

$$\text{place of service: } \mathbf{x}_s = (\underbrace{s_1, \cdots, s_1}_{n_1 \text{ count}}, \cdots, \underbrace{s_K, \cdots, s_K}_{n_K \text{ count}}), \quad \text{age: } \mathbf{x}_a = (a_1, \cdots, a_N),$$

$$\text{visit number: } \mathbf{x}_v = (\underbrace{1, \cdots, 1}_{n_1 \text{ count}}, \cdots, \underbrace{K, \cdots, K}_{n_K \text{ count}}), \quad \text{position: } \mathbf{x}_p = (1, \cdots, N),$$

where $\mathbf{x}_p$ corresponds to the position of codes in the sequence. Therefore, we represent a patient $X = \{\mathbf{x}_c, \mathbf{x}_s, \mathbf{x}_v, \mathbf{x}_a, \mathbf{x}_p\}$ as a collection of these sequences describing the medical record. Note that the record is organized chronologically with random ordering of the codes inside a visit. The tokenized code sequence is prepended with demographic tokens including a token $c_a$ for the patient age (in years, binned into 5-year age groups – e.g., 20-24, 25-29, etc.) of the patient and a token $c_s$ for patient sex (male or female): $\mathbf{x}_c = (c_a, c_s, c_{11}, \ldots, c_{Kn_K})$. Other context information sequences are also padded at the beginning accordingly. We utilize five different embedding layers to construct the final input sequence to the transformer model: (i) code embeddings $W_c \in \mathbb{R}^{\text{—vocab—} \times m}$, (ii) visit embeddings $W_v \in \mathbb{R}^{\text{max visit size} \times m}$, (iii) place-of-service embeddings $W_s \in \mathbb{R}^{|pos| \times m}$, (iv) time/age embeddings $W_a \in \mathbb{R}^{\text{max age} \times m}$ and finally, (v) standard positional embeddings $W_p \in \mathbb{R}^{\text{max seq length} \times m}$, where $m$ is the embedding size. Each element of the padded $\mathbf{x}_c, \mathbf{x}_s, \mathbf{x}_v, \mathbf{x}_a$, and $\mathbf{x}_p$ sequences are then one-hot encoded to the desired dimensions —vocab—, —pos—, max visit size, max age, max seq length, and passed through the embedding layers. The output of the embedding layers are then added up together to construct the input to the CHIRon transformer model.

**Visit-based causal masking:** In addition to the traditional causal masking for training CHIRon, we introduce a masking approach based on patient's visits. In this approach, each code in the current visit can only attend to the codes that occurred in the visits prior to the current visit. The attention masks in this case will be custom 2-D matrices created based on the visit number vector. Figure 9 in the appendix compares a regular causal attention mask and visit-based attention mask for a given example of a visit number sequence.

**Additional task heads for predicting place-of-service and age-at-encounter:** By adding two extra task heads to the model we enable prediction of next age-at-encounter and place-of-service for each token. The final loss is a weighted sum of the CE loss for code prediction and the CE losses for these extra heads. We used fixed weighting based on initial loss value to balance these losses during pretraining. The pretrained CHIRon model with the extra heads is referred to as CHIRon+ in the rest of this paper.

**Architecture and Hyperparameters for Pre-Training (PT):** We implemented the CHIRon architecture using the HuggingFace transformers (Wolf et al., 2020) package (v.4.25.1) and Pytorch (Paszke et al., 2019) (v2.0.1). The model contains a total of 6,392,832 parameters. For the transformer architecture of the CHIRon we used 6 layers, 8 heads, and embedding dimensionality of 256. The maximum sequence length is set to 512 and the inner feed forward layers have a dimension of 512. We also used the default attention dropout ratio and initializer range. We used the AdamWeight decay optimizer (Loshchilov & Hutter, 2019) with coefficient 0.01 and trained the model for 5e6 steps with early stopping of patience 3 using 2 Nvidia Tesla V100 GPUs.

**Architecture and Hyperparameters for Fine-Tuning (FT):** In fine-tuning, the code sequence is appended with a [CLS] token to use for classification. Our disease onset classification tasks are binary classification and we put a logistic FFL prediction head on top of the final layer of CHIRon. The fine-tuning transformer architecture is similar to the pretrain model. Starting from the pre-trained model, we train a separate model for each condition for 20 epochs with early stopping of patience 3 and batch size 64 using a single Nvidia Tesla V100 GPU.

### 3.3 DISEASE CLASSIFICATION

The pre-trained CHIRon model is fine-tuned for five separate binary classification tasks: predicting disease onset for chronic kidney disease (CKD), chronic obstructive pulmonary disease (COPD), dementia, diabetes, and predicting CKD disease progression (CKD-P, from stage 1-3a to stage 3b+). Cohort creation details can be found in Appendix A. Classification cohort sizes ranged from 382k (CKD-P) to 3.3M (diabetes) individuals (see Appendix Table 4). To fine-tune the model, we append a classification (CLS) token to the code sequence and add a feed-forward neural network layer on top of the final layer's classification token embedding. During fine-tuning, we allow the entire model to be updated.

For comparison, we used state-of-the-art and other common classification methods: gradient-boosted trees (GBT), RETAIN (Choi et al., 2017), Med-BERT (Rasmy et al., 2021), and TransformEHR (Yang et al., 2023). While Med-BERT and TransformEHR originally only used diagnosis codes, we included procedure codes and medications as input to the Med-BERT and TransformEHR model (plus lab codes for TransformEHR) for a more fair comparison, and denote this with "Med-BERT*" and "TransformEHR*" (see Appendix C for Med-BERT results using only diagnosis codes).

### 3.4 SEQUENTIAL MEDICAL DATA GENERATION

We used the HuggingFace transformers (Wolf et al., 2020) *model.generate()* function for auto-regressive generation of new codes with the base CHIRon model. We used beam search with *num_beams* = 5 and *do_sample* = *True* for generation and suppressed the rare codes "[CODE_TYPE_RARE_CODE]". The generation of new codes takes place one code at a time, i.e. the generated code at time $t$ is used in the sequence for the generation of the code at time $t + 1$. We also make use of the other additional context information in the generation process and pad them at each time step: place of service is padded with the unknown token and the other sequences such as visit number and patient age at encounter are padded with their most recent value.

With the CHIRon+ model, we are able to also generate the next place-of-service and age-at-encounter alongside the medical code. We use a modified version of the *model.generate()* for the CHIRon+ that outputs the generated place-of-service and age-at-encounter information and use them as additional context for generating the future codes. In this case, the visit number sequence is padded with its most recent value unless the generated age-at-encounter value changes.

To estimate the generative performance of CHIRon, we use a truncation procedure to remove medical codes from the end of a patient record, and evaluate how similar the generated codes are to the truncated codes. Specifically, we filter the pre-training validation set to select patients who have at least 50 codes. We truncate the last (most recent) $T$ codes from each record, and these $T$ codes are used as our reference (ground truth) code sequences. Using the truncated records as input to the model, the CHIRon model generates $T$ additional codes for each record. We then compare the reference code sequences with the generated code sequences to determine model performance for this generative task. We empirically show that adding the additional task heads for place-of-services and age-at-encounter prediction during pretraining improves the generation capabilities of CHIRon.

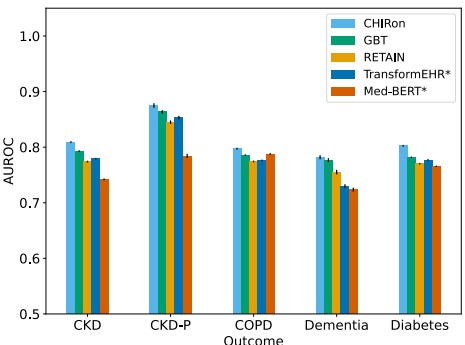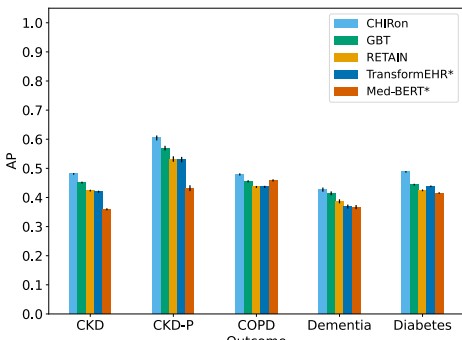

Figure 2: Classification performance in terms of (a) area under the ROC curve (AUROC) and (b) average precision (AP, or area under the precision-recall (PR) curve) for each model across all disease outcomes. Error bars indicate bootstrapped 95% confidence intervals.

## 4 EXPERIMENTS

### 4.1 DISEASE CLASSIFICATION

The pre-trained CHIRon model was fine-tuned for five binary classification tasks and compared with the baseline models. Figures 2 compare area under the ROC curve (AUROC) and average precision (AP) metrics for all models across the five tasks. In four out of five classification tasks, the GBT models were the strongest baseline, consistently outperforming both RETAIN and Med-BERT* in terms of both AUROC and AP by a statistically significant difference. The fine-tuned CHIRon model achieved the highest AUROC and AP in four of the five classification tasks (CKD, CKD-P, COPD, diabetes) by a statistically significant margin, and did as well as the GBT model in the fifth task (dementia). See Appendix Tables 5, 6, 7, 8 and 9 for numeric results.

### 4.2 EMBEDDING CONTINUOUS LAB RESULTS

We evaluated several methods for embedding continuous lab results into the CHIRon model: (1) tokenization per lab code per decile/percentile-range bin, (2) tokenization per lab code per bin with embedding scaling, and (3) tokenization per lab code and with shared decile embeddings, and measured their effect on the downstream disease onset classification tasks. In Figure 3 we show the AUROC and the AP for each lab embedding method across all disease outcomes. The results show that (1) across all conditions the decile embedding method outperformed the embedding scaling and the shared decile embedding method. In 4 out of 5 conditions this is statistically significant. (2) Using the scaled embeddings for each lab token indeed resulted in improved performance over these tasks, however this was only statistically significant for CKD, diabetes and COPD. This is particularly promising given that in the CKD and diabetes outcomes, lab results are expected to be a strong predictor of disease onset. (3) Using percentile range bins given in Table 15 did not improve the performance of the model on these downstream classification tasks. And finally, (4) adding a shared decile embedding vector to the original lab code embedding without partitioning for each decile/percentile-range bin had the lowest performance across all conditions. Though for dementia this was not significant compared to two of the baselines. See Appendix Tables 16, 17, 18, 19 and 20 for numeric results.

### 4.3 ADDITIONAL TASK HEADS AND VISIT-BASED CAUSAL MASKING

We pre-trained the CHIRon base model using the visit-based causal masking approach described in Sec 3.2 with our pre-training cohort and fine-tuned the model using our five disease classification cohorts. We also did the same experiment with CHIRon+ model – the base model with additional task heads for predicting place-of-service and age-at-encounter information. The results are presented in Figure 4. Interestingly, using visit-based attention masking deteriorated the performance of the fine-tuned models on all of the conditions. This empirically indicates that even though having access to the codes that previously occurred in the current visit could be considered a form of leakage during

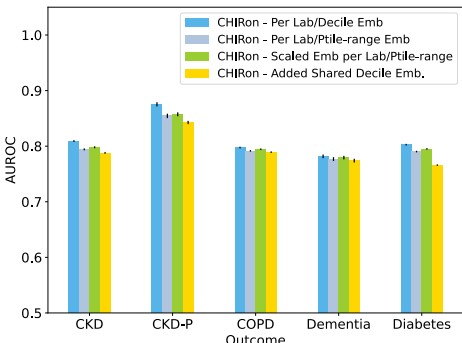 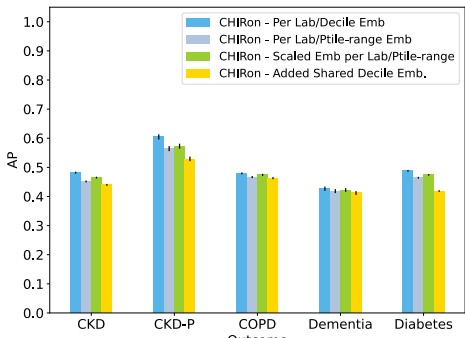

Figure 3: Classification performance in terms of (a) area under the ROC curve (AUROC) and (b) average precision (AP, or area under the precision-recall (PR) curve) for each model across all disease outcomes. Error bars indicate bootstrapped 95% confidence intervals.

the pre-training, it helps in learning better representations. It is important to note that since no future information is used by the "[CLS]" token, this does not result in leakage on the finetuning tasks used for evaluation. The results on the performance of the CHIRon+ model shows no statistically significant decline over the base CHIRon other than the CKD models. However, as we will see later, the CHIRon+ model empirically performs better in conditional sequential code generation tasks. We also provide the performance results for CHIRon+ using the scaled embeddings for lab codes in Figure 10, Appendix C. See Appendix Tables 21, 22, 23, 24 and 25 for numeric results.

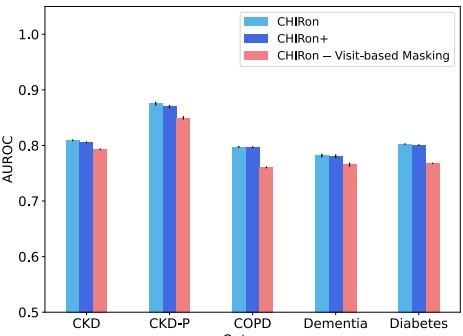 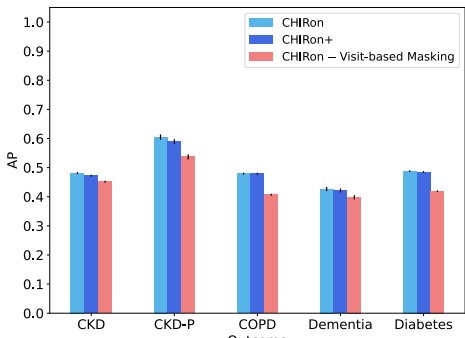

Figure 4: Classification performance in terms of (a) area under the ROC curve (AUROC) and (b) average precision (AP, or area under the precision-recall (PR) curve) for each model across all disease outcomes. Error bars indicate bootstrapped 95% confidence intervals.

### 4.4 SEQUENTIAL MEDICAL DATA GENERATION

Just as generative models for text can be used to generate synthetic text sequences based on an initial prompt, we can similarly generate synthetic sequential medical data. For a given medical record, we can use the generative capabilities of the pre-trained CHIRon model to sample additional synthetic patient data.

To quantitatively evaluate the generative performance, we adopt two established metrics from the NLP community: the ROUGE (Lin, 2004) score and the BERTScore (Zhang et al., 2020). The ROUGE-1 score measures the overlap of unigrams (single words/codes) between the reference sequence and the generated sequence. The BERTScore is a method for computing the similarity between two sequences as the mean cosine similarity between contextualized embeddings from the reference sequence and the generated sequences. Compared to the ROUGE score, the BERTScore penalizes a model less for generating codes that are very similar terms of medical taxonomy but not exact matches – as an example, if the model generates an ICD code "S92.812A" for a fracture of the left

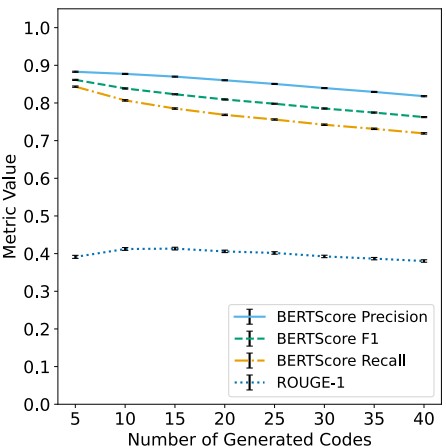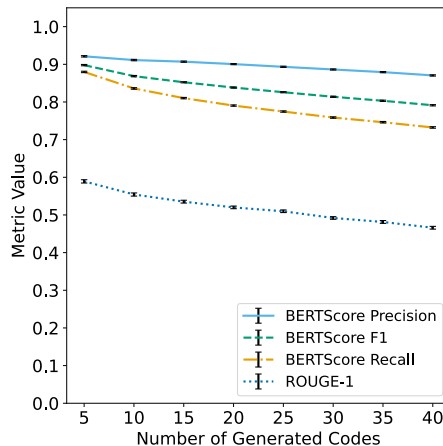

Figure 5: BERTScore and ROUGE metrics for (a, left) **CHIRon** and (b, right) **CHIRon+** as a function of the number of truncated/generated codes. Error bars indicate bootstrapped 95% confidence intervals.

foot versus "S92.901A" for a fracture of the right foot. Using these two metrics allows us to quantify how well the model can generate medical codes both exactly and semantically.

In Figure 5, we show ROUGE scores and BERTScore metrics for CHIRon and CHIRon+ as we vary the number of truncated codes. To calculate the metrics in this evaluation, we used 10,000 patient records and the same truncated sequences were put into both models at each time step. Notably, for both models the code generation is more precise than it is sensitive. We find that the accuracy of the generated codes decreases as we truncate more codes from the record. This is expected – as more codes are truncated, more context is removed. In general, it is also more difficult to predict codes that occur farther in the future. Overall, CHIRon+ has both higher BERTScore and higher ROUGE-1 score across the provided range of truncated codes compared to the base CHIRon. Numeric performance metrics can be found in Appendix Tables 26 and 27.

As the CHIRon+ model also enables predicting place-of-service and age-at-encounter alongside the codes, we evaluated the performance of the model on generating the sequences of place-of-service and age-at-encounter in a similar manner. For more details see Appendix D.

Figure 6 shows a BERTScore cosine similarity heatmap comparing contextualized embeddings from true and generated codes for an example patient using the CHIRon. BERTScore precision searches for the highest similarity in each row, whereas BERTScore recall searches for the highest similarity in each column. As expected, the similarity is highest when the codes are an exact match. However, because the codes are contextualized, the same code at two different positions in the sequence can have different embeddings (and therefore different similarity to the query code).

## 5   DISCUSSION

In this work we developed CHIRon and showed that given sequential medical data, the model is able to effectively generate realistic synthetic sequences of additional medical codes. Additionally, we found that fine-tuning this model for disease onset prediction achieves the best classification results in four of the five outcomes compared to four strong baseline methods.

**CHIRon as a foundation model:** We consider CHIRon a foundation model because it provides a robust base that generalizes to a diverse set of disease classifification tasks without any task-specific modifications. It achieves this by (i) having state-of-the-art performance on 5 distinct disease onset/prediction tasks and (ii) offers realistic generation capabilities. We note that previous foundation models for structured sequential medical data, e.g. Med-BERT, either only satisfy (i), or they are evaluated on fewer tasks (2 vs. 5). While generalization to many different datasets may be an expectation for some text-based foundation models, previous foundation models for structured

sequential medical data were not evaluated using many such datasets, as access to high quality large datasets of structured sequential medical data is typically limited. For example, both CHIRon and Med-BERT are evaluated using two such datasets. We note, however, that the two datasets CHIRon is trained on represent the largest amount of patient data used to train a foundation model for structured sequential medical data to our knowledge.

**Use of large private clinical/claims datasets rather than well-known public MIMIC-IV dataset:** While we agree MIMIC-IV is a well-known public resource for high quality patient data from ICU stays, both CHIRon and related foundational models focus on more general longitudinal healthcare data, not specific to ICU stays. Since, to our knowledge, there are no significant, high quality public sources of such data, both CHIRon and related approaches like Med-BERT use large private datasets. While we agree there is a disadvantage from these datasets not being public, we believe they are a better data source to use for pre-training since (i) they are much larger than MIMIC-IV, (ii) they offer greater diversity in the type of healthcare data included since it is not only from ICU stays, and (iii) they are more consistent with the data used originally used to evaluate the baselines we compare CHIRon against.

**CHIRon generation capabilities and full synthetic data generation:** Our experiments show that CHIRon demonstrates strong generative per-

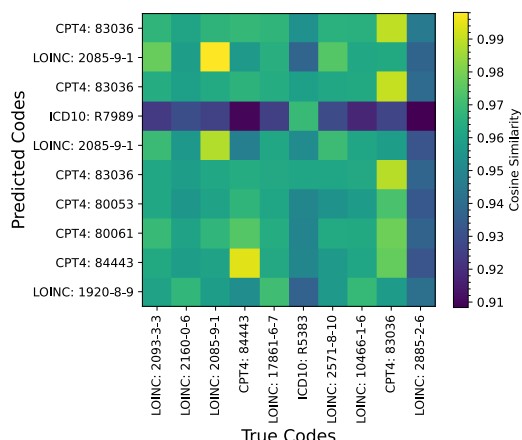

Figure 6: BERTScore cosine similarity heatmap comparing contextualized embeddings for $T = 10$ true (x-axis) and predicted (y-axis) codes from an example patient.

formance when used for conditional generation of additional codes given some existing patient history. We believe this capability has use cases in precision medicine since these additional codes could signal possible future clinical events. We believe unconditional generation of full synthetic patient records is another interesting but different use case which requires additional research to ensure the generated sequences represent the full diversity of patients included in the training set. One approach we are considering is leveraging the CHIRon pretrained model with other unconditional synthetic data generation approaches like diffusion models, but this alone is its own research topic.

We developed and evaluated several methods for embedding continuous lab values into the model. Incorporating the continuous values into FMs typically poses a significant challenge due to the variability in both scale and distribution of different lab tests. This makes it especially difficult to also standardize the values across different lab tests for modeling. Previous work such as Golkar et al. (2023) saw better improvements by scaling the embeddings. However, the dataset used in that work was more balanced in terms of having numeric vs non-numeric tokens. While we specifically focused on lab results as our continuous data type, this problem is more general, and this method can be extended to other continuous data types such as vital signs or medication dosage. By adding extra tasks for predicting the place-of-service and age-at-encounter information alongside the next code, we improved the generative performance of CHIRon. Our experiments with the visit-based attention masking approach show that having access to previously predicted codes inside each visit can improve the learned representations during pre-training.

We note several limitations. We trained and validated the models on data from a single institution; in future work, we hope to validate model generalization to external datasets for both the generation and classification tasks. Additionally, due to the expensive nature of training large FMs, we were not able to conduct extensive hyperparameter tuning – results may improve with further investigation. Future work can explore the combination of synthetic sequential patient data generation with the classification task: by generating additional codes and better estimating a patient's trajectory, an augmented patient record may improve downstream classification performance.

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
