## A  DATASETS AND COHORT SELECTION

**Datasets:**  a large healthcare institution's de-identified data consists of de-identified administrative claims data such as medical and pharmacy claims, laboratory (labs) results, enrollment records, and demographic (age and sex only) information. Our cohorts included individuals with records occurring between 1 January 2014 and 31 December 2018. Individuals younger than 18 years or older than 89 years were excluded from our analysis. For all individuals we extracted a list of all medical codes with their corresponding timestamp and place-of-service information, and ordered the codes chronologically.

Diagnoses were coded using the International Classification of Diseases (ICD), Ninth and Tenth Revisions, Clinical Modification (ICD-9-CM and ICD-10-CM). Any ICD-9 codes were mapped ICD-10. Procedures were coded using the Current Procedural Terminology (CPT-4). For lab results, we use the Logical Observation Identifier Names and Codes (LOINC) standardized system to identify specific lab tests. Medications were grouped by generic name. American Medical Association (AMA) place-of-service code information for each medical code was categorized as one of these six categories: outpatient, inpatient, emergency, custodial, independent lab, home, or otherwise unknown.

The structured healthcare records were stored in a table with the following information:

- id: unique patient identifier
- enc_dt: date of encounter
- code_type: type of code - e.g., diagnosis (DIAG_ICD10), procedure (PROC_CPT4), etc.
- code: diagnosis (e.g., N18.1), procedure (e.g., 99214), etc.
- pos: place of service code

**Pre-training cohort:**  We combined both the claims and clinical datasets into a single large dataset to pre-train the transformer-based models. We required that individuals be enrolled for 3 or more months, and have 6 or more days with encounters (diagnosis, procedure, lab, medication fill record) to be included. Identical records from claims and clinical data datasets were de-duplicated. This combined dataset included over 13.88 billion medical codes from 44,169,102 unique individuals. See Table 1 for counts of individuals, total number of ICD codes, and unique number of ICD codes for each dataset in our pre-training cohort.

Table 1: For each dataset we report the number of unique individuals, number of total ICD codes, number of unique ICD codes, the median (IQR) number of codes per each individual, the median (IQR) number of encounters per each individual, the median (IQR) age in years, and the percent female.

| Dataset | Individuals | Total records | Unique codes | Codes per patient | Encounters per patient | Age (years) | Sex (%F) |
|---|---|---|---|---|---|---|---|
| Claims | 21,493,605 | 6.55B | 7,615 | 164 (72-386) | 35 (15-81) | 51 (35-64) | 57.3 |
| EHR | 25,669,520 | 7.52B | 7,261 | 151 (69-337) | 19 (9-42) | 50 (34-63) | 59.8 |
| Combined | 44,169,102 | 13.88B | 7,922 | 165 (73-378) | 26 (12-61) | 51 (34-64) | 58.5 |

**Fine-tuning cohorts:**  For the classification cohorts used for fine-tuning the CHIRon model and method comparison, we again leveraged the combined claims and clinical datasets. In the classification, we required that individuals be enrolled for a minimum of two years to ensure a minimum time window for outcomes to occur, and have 6 or more days with encounters.

Cases (positive labels) were defined as individuals who had at least two occurrences of any inclusion code (see Table 2) between 7-365 days apart in any setting (inpatient, outpatient, etc.), or one occurrence of a diagnosis code (see Table 2) in an inpatient setting (inpatient setting is more reliable). We used the earliest occurrence of any inclusion code as the "index date".

We excluded individuals if they met our exclusionary criteria: for CKD/CKD-P, if individuals had a diagnosis code for acute and unspecified renal failure, other specified and unspecified diseases of

Table 2: Medical codes used to define classification cohort cases. Case definition required at occurrence of at least two of these codes within a 365-day period. For lab values, we report the LOINC code and the lab result range that would count as evidence of the outcome (e.g., a LOINC 4548-4 ("Hemoglobin A1c/Hemoglobin.total in Blood") result between 6.501% and 100% is evidence of diabetes). ICD: International Classification of Diseases; CCSR: Clinical Classifications Software Refined; ETG: Episode Treatment Group; CPT-4: Current Procedural Terminology; PCC: ; LOINC: Logical Observation Identifier Names and Codes.

| Outcome | ICD | CCSR | ETG | CPT-4 | PCC | LOINC (min, max) lab range |
|---|---|---|---|---|---|---|
| CKD | E11.21, E11.29, N08 | GEN003 | | | | 33914-3 (0,75), 62238-1 (0,75), 98979-8 (0,75) |
| CKD-P | N18.32, N18.3, N18.5, N18.6 | | | | | 33914-3 (0,45), 62238-1 (0,45), 98979-8 (0,45) |
| COPD | | RSP008 | | | 574, 576, 577, 605 | |
| Dementia | | | 239000, 316400 | 99307, 99308, 99309, 99310 | 344 | |
| Diabetes | | END003, END005 | | | 500-504, 513-515, 518, 520, 523, 841 | 1558-6 (126,1200), 2339-0 (150,1200), 2345-7 (150,1200), 27353-2 (150,1200), 4548-4 (6.501,100) |

kidney and ureters, or kidney transplant status; for dementia, if individuals had a procedure code indicating they were living in a skilled nursing facility or hospice; for diabetes if an individual had any diagnosis codes that indicated Type-1 diabetes (see Table 3 for codes used).

Table 3: Medical codes used to exclude individuals from our classification cohorts. Any occurrence of these codes would exclude an individual from being a case or control for each outcome. ICD: International Classification of Diseases; CCSR: Clinical Classifications Software Refined; ETG: Episode Treatment Group; CPT-4: Current Procedural Terminology; PCC: ; LOINC: Logical Observation Identifier Names and Codes.

| Outcome | ICD | CCSR | ETG | CPT-4 | PCC | LOINC (min, max) lab range |
|---|---|---|---|---|---|---|
| CKD | Z94.0 | GEN002, GEN006 | | | | |
| CKD-P | Z94.0 | GEN002, GEN006 | | | | |
| COPD | | | | | | |
| Dementia | | | | 99307, 99308, 99309, 99310 | | |
| Diabetes | | END004 | | | | |

We defined controls (negative labels) as individuals who had neither an inclusion code nor an exclusion code over the entire period for which we have their clinical history. Controls were randomly selected so that the final cohort had a case:control ratio of 1:5 (i.e., a 16.67% prevalence). Final cohort counts can be found in Table 4.

The cohorts were split into train and test using a time-based split, such that individuals with the most recent 20% were included in the test set and the remaining (earliest) 80% were used for model training. The training set was further randomly split into training and validation sets, such that 70% of the total data was used for model training and the remaining 10% was used for validation.

Table 4: Classification cohort sizes and case counts. Using our case/control sampling procedure, each cohort maintains a 1:5 case:control ratio (i.e., a prevalence of 16.67%).

| Outcome | Individuals | Number of cases |
|---|---|---|
| CKD | 2,427,678 | 404,613 |
| CKD-P | 382,632 | 63,772 |
| COPD | 3,197,016 | 532,836 |
| Dementia | 583,896 | 97,316 |
| Diabetes | 3,299,352 | 549,892 |

## B  MODEL COMPARISONS

**Gradient-boosted trees (GBT)**  Gradient boosted trees (GBTs) which modeled the presence of codes, and not the sequence in which they occurred, were trained as one of the baseline models. Each cell in the model matrix was populated with the number of days a patient had a specific code. The demographic information was added using binary columns. A threshold of 0.0001 was set for initial feature selection – i.e., a code should appear in at least 10,000 patients to be included in the initial feature set. A two step approach was implemented for tuning the models. In the first step 150 Optuna trials were run and the features which had at least 1 split in 20% of those trials were selected for the second step. In the second step 300 Optuna trials were run on the pruned set of features. Average precision was the measure selected for early stopping the boosted tree training runs and Optuna trial evaluation. Libraries used for this comparison included lightgbm (Ke et al., 2017) (v3.3.5) and Optuna (Akiba et al., 2019) (v3.1.0) for hyperparameter tuning.

**RETAIN**  RETAIN (Choi et al., 2016) is an interpretable two-level neural attention predictive model with applications to EHR data. It considers both the visit-level and the variable-level influence of each visit through two sets of attentions weight. It generates the attention vectors by running two RNNs backward in time and then creates the final context vector to use for classification. For more details on the model architecture see Choi et al. (2016). We used a Pytorch-lightning implementation of the original RETAIN code[2] based on modifications done in this repo [3]. We included the place of service, visit number, and time/age at encounter as well as demographic information as additional features in the model. For each condition, we trained a separate model for each of the three age brackets and data sources and aggregated the results in Figure 7. We did a coarse grid search hyperparameter tuning of the diabetes model and used the parameters for the rest of the RETAIN models.

**Med-BERT**  Med-BERT (Rasmy et al., 2021) is a BERT-based model that, instead of using text as input, learns contextualized representations of sequentially-ordered medical codes. Similar to BERT (Devlin et al., 2019) models, Med-BERT can be pre-trained using an objective like masked language modeling on a large, general cohort of patients, and then fine-tuned on a smaller, more specific cohort for downstream tasks, e.g., disease onset prediction.

We implemented the Med-BERT architecture using the HuggingFace transformers (Wolf et al., 2020) package (v.4.25.1) and Pytorch (Paszke et al., 2019) (v2.0.1) using the same hyperparameters as described in Rasmy et al. (2021). We then pre-trained a Med-BERT model from scratch using the

---

[2] https://github.com/mp2893/retain
[3] https://github.com/ast0414/pytorch-retain

same preprocessing procedure such as not using the [CLS] token for input representation as described in Rasmy et al. (2021). We used the same tokenizer as the CHIRon model (described above in 3.1).

There are a few differences between the original Med-BERT (Rasmy et al., 2021) training and the Med-BERT training results presented in this paper: (1) We only pre-train the Med-BERT on the masked language modeling task. The original Med-BERT paper also includes a "prediction of prolonged length of stay in hospital" task. To perform this, the authors only include *inpatient* admissions in their data. However, we consider *all* types available encounters. (2) The original Med-BERT paper uses only diagnosis codes. To provide a fair comparison between this model and our CHIRon model, we trained two sets of Med-BERT models, one using only the diagnosis codes (ICD-10) in our data (denoted "Med-BERT"), and one with also including the procedure and medication codes (denoted "Med-BERT*"). Comparisons of the two models is included in Figures 7 and 7 and Tables 10, 11, 12, 13, and 14.

To fine-tune the Med-BERT models, we attached a classification head to the pre-trained Med-BERT model using the "Med-BERT_only (FFL)" head, in which a feed-forward layer (FFL) is added on top of the first token's embedding as in the original Med-BERT implementation [4]. We fine-tuned the model using a batch size of 64 for a maximum of 20 epochs, with early stopping if the model's validation loss does not decrease after 3 epochs.

The pre-training and fine-tuning cohorts that were used are the same as the cohorts used for CHIRon.

**TransformEHR**  TransformEHR (Yang et al., 2023) is likely the closest model to CHIRon in the previous literature. It is an encoder-decoder model that uses a BART-style architecture. The original model is trained only on ICD-10 codes and utilizes embeddings for visit number, patient demographics and time in the form of the number of days from patient's last visit. The original pre-training cohort includes regularly-sampled EHR data from 6.5 million patients from the MIMIC-IV dataset and the model is later fine-tuned for disease or outcome prediction tasks.

The complete implementation of TransformEHR (Yang et al., 2023) (including the code for pre-training) is not publicly available. Therefore we implemented the TransformEHR architecture using the HuggingFace transformers (Wolf et al., 2020) package (v.4.25.1) and Pytorch (Paszke et al., 2019) (v2.0.1) using the hyperparameters described in Yang et al. (2023) and pre-trained the model from scratch using our pre-training cohort. Similar to the approach in Yang et al. (2023) the model was pre-trained on predicting the codes in the next visit with a causal decoder. For a fair comparison with our CHIRon model, the data that we used to pre-train TransformEHR also includes the procedure, medication, and lab codes. Similar to the original paper, we also used the number of days from patient's last encounter as the input to its temporal embedding layer. Our TransformEHR pre-training is done on predicting the codes in the patient's last visit, given all the codes from the previous visits. (A pre-training on predicting the codes in all visits for each patient would require a large amount of data augmentation for CHIRon). We then fine-tuned TransformEHR for the same disease onset and progression tasks. The fine-tuning cohorts that were used are the same as the cohorts used for CHIRon.

---

[4] https://github.com/ZhiGroup/Med-BERT

## C EVALUATION RESULTS

In this section we provide numerical values for figures provided in the Sec 4 and results on some additional experiments. As mentioned in Sec B, TransformEHR uses the number of days from patient's last visit as the time input to the temporal embeddings in their model. We did a similar experiment with CHIRon and instead of age-at-encounter in months we used the number of days from the last visit. Figure 7 shows the calculated metrics for CHIRon traned using this information. As it is shown the performance is not statistically significantly different from a CHIRon model trained on age-at-encounter in months across all out disease classification tasks. See Tables 10, 11, 12, 13, and 14 for numerical values.

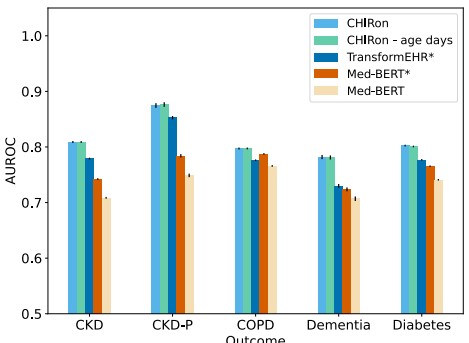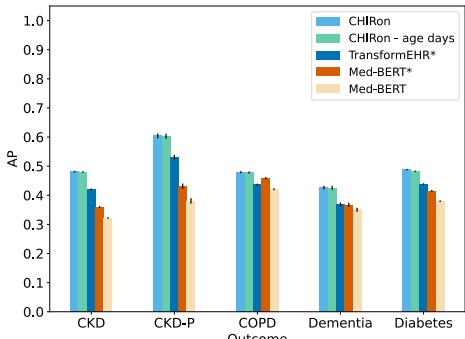

Figure 7: Classification performance in terms of (a) area under the ROC curve (AUROC) and (b) average precision (AP, or area under the precision-recall (PR) curve) for each model across all disease outcomes. Error bars indicate bootstrapped 95% confidence intervals.

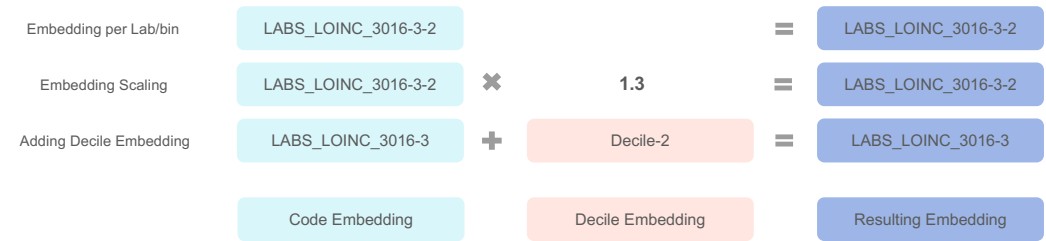

Figure 8: Three methods for embedding continuous lab results: (top) tokenization per lab code per bin with embedding scaling based on the (scaled) continuous lab value, (middle) tokenization per lab code and adding decile-specific tokens, and (bottom) tokenization per lab code per decile bin,

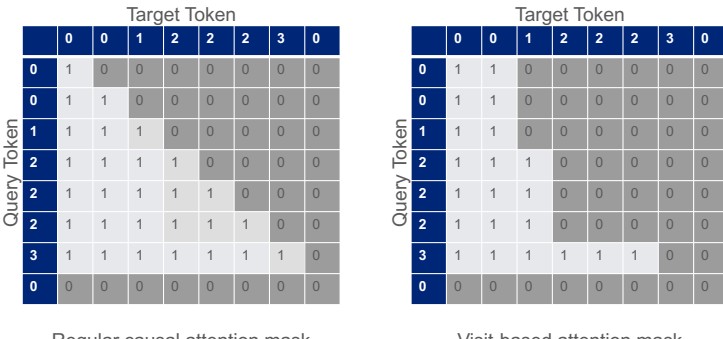

Figure 9: comparing regular causal attention mask and visit-based attention mask for visit ID sequence: [0,0,1,2,2,2,3,0]. Note that in this case patient has 3 visits and the last one is the pad ID.

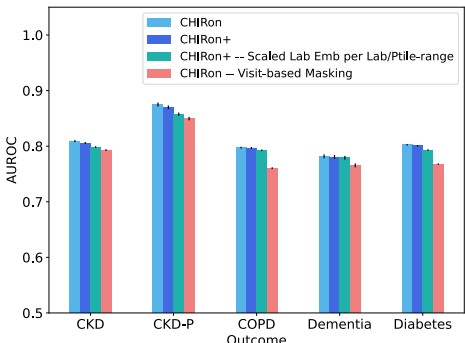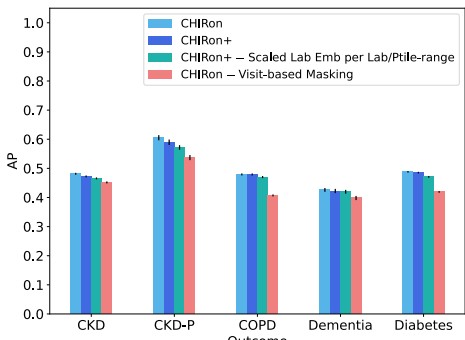

Figure 10: Classification performance in terms of (a) area under the ROC curve (AUROC) and (b) average precision (AP, or area under the precision-recall (PR) curve) for each model across all disease outcomes. Error bars indicate bootstrapped 95% confidence intervals.

## D SEQUENTIAL DATA GENERATION

As CHIRon+ model also enables predicting place-of-service and age-at-encounter alongside the codes, we evaluated the performance of the model on generating the sequences of place-of-service and age-at-encounter in a similar manner. However, instead of calculating the cosine similaty between the contextualized embeddings, we simply used the uncontextualized embeddings. See Figure 11, Tables 28, 29.

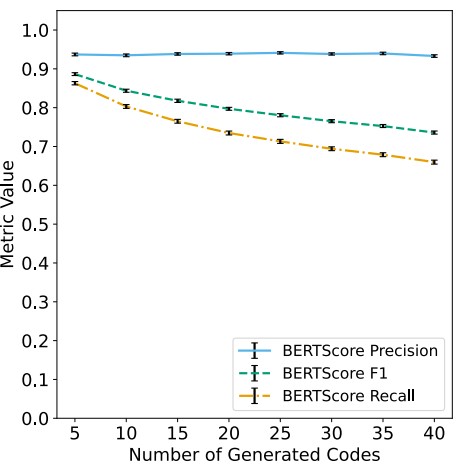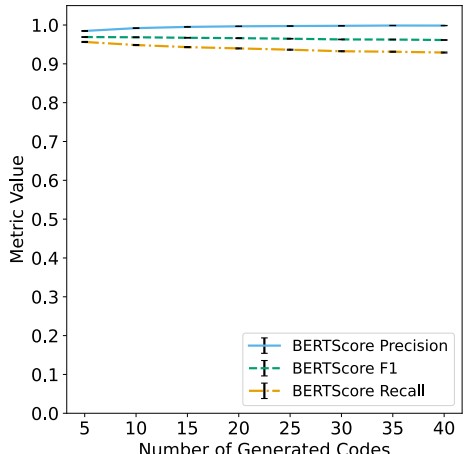

Figure 11: Left: **Age-at-encounter**:, right: **Place-of-service**. Generative performance metrics as a function of the number of codes generated. The BERTScore F1 metric is calculated between the uncontextualized generated and true sequences. Error bars indicate bootstrapped 95% confidence intervals.

Table 5: Classification performance metrics for CKD

| metric model | AUROC | AP |
|---|---|---|
| CHIRon | 0.809 (0.808-0.811) | 0.481 (0.478-0.485) |
| GBT | 0.793 (0.791-0.795) | 0.451 (0.448-0.454) |
| RETAIN | 0.774 (0.772-0.776) | 0.423 (0.420-0.427) |
| TransformEHR* | 0.779 (0.778-0.781) | 0.420 (0.417-0.423) |
| Med-BERT* | 0.742 (0.740-0.744) | 0.359 (0.357-0.362) |

Table 6: Classification performance metrics for CKD-P

| metric
model | AUROC | AP |
|---|---|---|
| CHIRon | 0.875 (0.872-0.878) | 0.605 (0.597-0.613) |
| GBT | 0.864 (0.861-0.867) | 0.569 (0.559-0.578) |
| RETAIN | 0.845 (0.841-0.849) | 0.532 (0.522-0.542) |
| TransformEHR* | 0.853 (0.850-0.856) | 0.530 (0.522-0.540) |
| Med-BERT* | 0.784 (0.780-0.788) | 0.431 (0.423-0.440) |

Table 7: Classification performance metrics for COPD

| metric
model | AUROC | AP |
|---|---|---|
| CHIRon | 0.797 (0.796-0.799) | 0.479 (0.475-0.483) |
| GBT | 0.786 (0.784-0.787) | 0.456 (0.452-0.459) |
| RETAIN | 0.774 (0.773-0.776) | 0.436 (0.432-0.440) |
| TransformEHR* | 0.776 (0.775-0.778) | 0.436 (0.433-0.440) |
| Med-BERT* | 0.787 (0.786-0.789) | 0.459 (0.455-0.462) |

Table 8: Classification performance metrics for Dementia

| metric
model | AUROC | AP |
|---|---|---|
| CHIRon | 0.782 (0.779-0.785) | 0.426 (0.419-0.432) |
| GBT | 0.777 (0.773-0.780) | 0.415 (0.407-0.421) |
| RETAIN | 0.755 (0.751-0.759) | 0.388 (0.381-0.395) |
| TransformEHR* | 0.730 (0.726-0.734) | 0.369 (0.362-0.376) |
| Med-BERT* | 0.724 (0.720-0.728) | 0.367 (0.360-0.375) |

Table 9: Classification performance metrics for Diabetes

| metric
model | AUROC | AP |
|---|---|---|
| CHIRon | 0.803 (0.801-0.804) | 0.488 (0.485-0.491) |
| GBT | 0.782 (0.781-0.783) | 0.444 (0.441-0.447) |
| RETAIN | 0.770 (0.769-0.772) | 0.424 (0.421-0.427) |
| TransformEHR* | 0.777 (0.776-0.778) | 0.439 (0.436-0.442) |
| Med-BERT* | 0.765 (0.764-0.767) | 0.415 (0.413-0.418) |

Table 10: Classification performance metrics for CKD

| metric
model | AUROC | AP |
|---|---|---|
| CHIRon | 0.809 (0.808-0.811) | 0.481 (0.478-0.485) |
| CHIRon - age days | 0.809 (0.807-0.811) | 0.480 (0.477-0.483) |
| TransformEHR* | 0.779 (0.778-0.781) | 0.420 (0.417-0.423) |
| Med-BERT* | 0.742 (0.740-0.744) | 0.359 (0.357-0.362) |
| Med-BERT | 0.709 (0.707-0.710) | 0.321 (0.318-0.324) |

Table 11: Classification performance metrics for CKD-P

| metric
model | AUROC | AP |
|---|---|---|
| CHIRon | 0.875 (0.872-0.878) | 0.605 (0.597-0.613) |
| CHIRon - age days | 0.876 (0.873-0.879) | 0.603 (0.595-0.612) |
| TransformEHR* | 0.853 (0.850-0.856) | 0.530 (0.522-0.540) |
| Med-BERT* | 0.784 (0.780-0.788) | 0.431 (0.423-0.440) |
| Med-BERT | 0.749 (0.744-0.753) | 0.380 (0.372-0.388) |

Table 12: Classification performance metrics for COPD

| metric
model | AUROC | AP |
|---|---|---|
| CHIRon | 0.797 (0.796-0.799) | 0.479 (0.475-0.483) |
| CHIRon - age days | 0.797 (0.796-0.799) | 0.478 (0.475-0.482) |
| TransformEHR* | 0.776 (0.775-0.778) | 0.436 (0.433-0.440) |
| Med-BERT* | 0.787 (0.786-0.789) | 0.459 (0.455-0.462) |
| Med-BERT | 0.766 (0.764-0.768) | 0.421 (0.417-0.424) |

Table 13: Classification performance metrics for Dementia

| metric
model | AUROC | AP |
|---|---|---|
| CHIRon | 0.782 (0.779-0.785) | 0.426 (0.419-0.432) |
| CHIRon - age days | 0.781 (0.777-0.784) | 0.425 (0.419-0.432) |
| TransformEHR* | 0.730 (0.726-0.734) | 0.369 (0.362-0.376) |
| Med-BERT* | 0.724 (0.720-0.728) | 0.367 (0.360-0.375) |
| Med-BERT | 0.707 (0.703-0.711) | 0.351 (0.345-0.357) |

Table 14: Classification performance metrics for Diabetes

| metric
model | AUROC | AP |
|---|---|---|
| CHIRon | 0.803 (0.801-0.804) | 0.488 (0.485-0.491) |
| CHIRon - age days | 0.801 (0.799-0.802) | 0.482 (0.479-0.485) |
| TransformEHR* | 0.777 (0.776-0.778) | 0.439 (0.436-0.442) |
| Med-BERT* | 0.765 (0.764-0.767) | 0.415 (0.413-0.418) |
| Med-BERT | 0.741 (0.739-0.742) | 0.379 (0.377-0.382) |

Table 15: Percentile bins for LOINC codes

| Number of records | Ptile bins breaking points |
|---|---|
| (100000, 100000000) | [0.001, 0.02, 0.05, 0.15, 0.3, 0.5, 0.7, 0.85, 0.95, 0.98, 0.999] |
| (20000, 100000) | [0.001, 0.05, 0.15, 0.3, 0.5, 0.7, 0.85, 0.95, 0.999] |
| (1000, 20000) | [0.001, 0.15, 0.3, 0.5, 0.7, 0.85, 0.999] |

Table 16: Classification performance metrics for CKD

| metric
model | AUROC | AP |
|---|---|---|
| CHIRon - Per Lab/Decile Emb | 0.809 (0.808-0.811) | 0.481 (0.478-0.485) |
| CHIRon - Per Lab/Ptile-range Emb | 0.794 (0.793-0.796) | 0.451 (0.448-0.455) |
| CHIRon - Scaled Emb per Lab/Ptile-range | 0.798 (0.796-0.800) | 0.465 (0.462-0.468) |
| CHIRon - Added Shared Decile Emb. | 0.788 (0.787-0.790) | 0.440 (0.437-0.444) |

Table 17: Classification performance metrics for CKD-P

| metric
model | AUROC | AP |
|---|---|---|
| CHIRon - Per Lab/Decile Emb | 0.875 (0.872-0.878) | 0.605 (0.597-0.613) |
| CHIRon - Per Lab/Ptile-range Emb | 0.855 (0.851-0.858) | 0.565 (0.556-0.573) |
| CHIRon - Scaled Emb per Lab/Ptile-range | 0.858 (0.854-0.861) | 0.572 (0.564-0.581) |
| CHIRon - Added Shared Decile Emb. | 0.843 (0.839-0.847) | 0.529 (0.520-0.538) |

Table 18: Classification performance metrics for COPD

| metric
model | AUROC | AP |
|---|---|---|
| CHIRon - Per Lab/Decile Emb | 0.797 (0.796-0.799) | 0.479 (0.475-0.483) |
| CHIRon - Per Lab/Ptile-range Emb | 0.791 (0.790-0.793) | 0.467 (0.463-0.470) |
| CHIRon - Scaled Emb per Lab/Ptile-range | 0.794 (0.793-0.796) | 0.474 (0.471-0.478) |
| CHIRon - Added Shared Decile Emb. | 0.789 (0.788-0.791) | 0.463 (0.460-0.467) |

Table 19: Classification performance metrics for Dementia

| metric
model | AUROC | AP |
|---|---|---|
| CHIRon - Per Lab/Decile Emb | 0.782 (0.779-0.785) | 0.426 (0.419-0.432) |
| CHIRon - Per Lab/Ptile-range Emb | 0.777 (0.773-0.780) | 0.419 (0.412-0.425) |
| CHIRon - Scaled Emb per Lab/Ptile-range | 0.780 (0.776-0.783) | 0.422 (0.416-0.429) |
| CHIRon - Added Shared Decile Emb. | 0.774 (0.771-0.777) | 0.412 (0.405-0.420) |

Table 20: Classification performance metrics for Diabetes

| metric
model | AUROC | AP |
|---|---|---|
| CHIRon - Per Lab/Decile Emb | 0.803 (0.801-0.804) | 0.488 (0.485-0.491) |
| CHIRon - Per Lab/Ptile-range Emb | 0.790 (0.789-0.792) | 0.464 (0.461-0.467) |
| CHIRon - Scaled Emb per Lab/Ptile-range | 0.795 (0.794-0.797) | 0.474 (0.471-0.477) |
| CHIRon - Added Shared Decile Emb. | 0.766 (0.765-0.768) | 0.419 (0.416-0.422) |

Table 21: Classification performance metrics for CKD

| metric
model | AUROC | AP |
|---|---|---|
| CHIRon | 0.809 (0.808-0.811) | 0.481 (0.478-0.485) |
| CHIRon+ | 0.806 (0.804-0.807) | 0.472 (0.469-0.476) |
| CHIRon+ – Scaled Lab Emb per Lab/Ptile-range | 0.798 (0.797-0.800) | 0.466 (0.462-0.469) |
| CHIRon – Visit-based Masking | 0.793 (0.791-0.795) | 0.452 (0.449-0.455) |

Table 22: Classification performance metrics for CKD-P

| metric model | AUROC | AP |
|---|---|---|
| CHIRon | 0.875 (0.872-0.878) | 0.605 (0.597-0.613) |
| CHIRon+ | 0.870 (0.867-0.873) | 0.589 (0.580-0.598) |
| CHIRon+ – Scaled Lab Emb per Lab/Ptile-range | 0.857 (0.854-0.861) | 0.571 (0.563-0.581) |
| CHIRon – Visit-based Masking | 0.849 (0.846-0.853) | 0.537 (0.528-0.547) |

Table 23: Classification performance metrics for COPD

| metric model | AUROC | AP |
|---|---|---|
| CHIRon | 0.797 (0.796-0.799) | 0.479 (0.475-0.483) |
| CHIRon+ | 0.797 (0.795-0.799) | 0.479 (0.476-0.482) |
| CHIRon+ – Scaled Lab Emb per Lab/Ptile-range | 0.792 (0.791-0.794) | 0.470 (0.467-0.474) |
| CHIRon – Visit-based Masking | 0.760 (0.759-0.762) | 0.407 (0.403-0.410) |

Table 24: Classification performance metrics for Dementia

| metric model | AUROC | AP |
|---|---|---|
| CHIRon | 0.782 (0.779-0.785) | 0.426 (0.419-0.432) |
| CHIRon+ | 0.781 (0.777-0.784) | 0.422 (0.414-0.429) |
| CHIRon+ – Scaled Lab Emb per Lab/Ptile-range | 0.779 (0.776-0.782) | 0.420 (0.413-0.426) |
| CHIRon – Visit-based Masking | 0.766 (0.762-0.769) | 0.399 (0.392-0.406) |

Table 25: Classification performance metrics for Diabetes

| metric model | AUROC | AP |
|---|---|---|
| CHIRon | 0.803 (0.801-0.804) | 0.488 (0.485-0.491) |
| CHIRon+ | 0.801 (0.799-0.802) | 0.485 (0.482-0.488) |
| CHIRon+ – Scaled Lab Emb per Lab/Ptile-range | 0.793 (0.792-0.795) | 0.471 (0.468-0.474) |
| CHIRon – Visit-based Masking | 0.768 (0.766-0.769) | 0.419 (0.416-0.422) |

Table 26: Generative performance metrics as a function of the number of codes generated for **CHIRon**. The BERTScore F1 metric is a measure of cosine similarity between the generated medical codes and the true medical codes. The ROUGE-$n$ score measures the overlap of code $n$-grams between the generated medical codes and the true medical codes. Error bars indicate bootstrapped 95% confidence intervals.

| NC | BERTScore F1 | BERTScore Precision | BERTScore Recall | ROUGE-1 |
|---|---|---|---|---|
| 5 | 0.861 (0.860-0.863) | 0.883 (0.881-0.884) | 0.843 (0.842-0.845) | 0.391 (0.387-0.395) |
| 10 | 0.839 (0.837-0.840) | 0.877 (0.876-0.879) | 0.807 (0.805-0.809) | 0.412 (0.409-0.416) |
| 15 | 0.823 (0.822-0.825) | 0.870 (0.869-0.871) | 0.785 (0.783-0.787) | 0.413 (0.410-0.417) |
| 20 | 0.809 (0.808-0.811) | 0.860 (0.859-0.862) | 0.768 (0.767-0.770) | 0.406 (0.403-0.409) |
| 25 | 0.798 (0.797-0.799) | 0.851 (0.849-0.852) | 0.756 (0.754-0.758) | 0.402 (0.398-0.405) |
| 30 | 0.785 (0.784-0.787) | 0.840 (0.838-0.841) | 0.742 (0.740-0.744) | 0.393 (0.389-0.396) |
| 35 | 0.774 (0.773-0.776) | 0.829 (0.828-0.831) | 0.731 (0.729-0.733) | 0.387 (0.383-0.390) |
| 40 | 0.762 (0.761-0.764) | 0.818 (0.816-0.820) | 0.719 (0.717-0.721) | 0.380 (0.377-0.384) |

Table 27: Generative performance metrics as a function of the number of codes generated for **CHIRon+**. The BERTScore F1 metric is a measure of cosine similarity between the generated medical codes and the true medical codes. The ROUGE-$n$ score measures the overlap of code $n$-grams between the generated medical codes and the true medical codes. Error bars indicate bootstrapped 95% confidence intervals.

| NC | BERTScore F1 | BERTScore Precision | BERTScore Recall | ROUGE-1 |
|---|---|---|---|---|
| 5 | 0.898 (0.896-0.900) | 0.921 (0.919-0.923) | 0.880 (0.878-0.882) | 0.589 (0.585-0.594) |
| 10 | 0.869 (0.867-0.870) | 0.911 (0.909-0.913) | 0.836 (0.834-0.838) | 0.554 (0.550-0.558) |
| 15 | 0.852 (0.851-0.854) | 0.907 (0.905-0.909) | 0.810 (0.808-0.812) | 0.535 (0.532-0.539) |
| 20 | 0.838 (0.837-0.840) | 0.901 (0.899-0.902) | 0.790 (0.789-0.792) | 0.520 (0.517-0.524) |
| 25 | 0.826 (0.824-0.827) | 0.893 (0.892-0.895) | 0.775 (0.773-0.776) | 0.510 (0.506-0.513) |
| 30 | 0.814 (0.812-0.815) | 0.886 (0.885-0.888) | 0.759 (0.757-0.761) | 0.492 (0.489-0.495) |
| 35 | 0.803 (0.802-0.805) | 0.879 (0.878-0.881) | 0.746 (0.744-0.748) | 0.481 (0.478-0.485) |
| 40 | 0.791 (0.790-0.793) | 0.871 (0.869-0.872) | 0.732 (0.730-0.734) | 0.466 (0.462-0.470) |

Table 28: Generative performance metrics as a function of the number of codes generated. The BERTScore F1 metric is a measure of cosine similarity between the uncontextualized generated and true **age-at-encounter** sequences. Error bars indicate bootstrapped 95% confidence intervals.

| Number of Codes | BERTScore F1 | BERTScore Precision | BERTScore Recall |
|---|---|---|---|
| 5 | 0.886 (0.883-0.890) | 0.937 (0.934-0.941) | 0.863 (0.859-0.867) |
| 10 | 0.844 (0.840-0.848) | 0.935 (0.931-0.938) | 0.803 (0.798-0.808) |
| 15 | 0.818 (0.813-0.822) | 0.938 (0.935-0.942) | 0.765 (0.760-0.770) |
| 20 | 0.797 (0.793-0.801) | 0.939 (0.936-0.942) | 0.735 (0.730-0.740) |
| 25 | 0.780 (0.777-0.785) | 0.941 (0.938-0.944) | 0.713 (0.708-0.718) |
| 30 | 0.765 (0.761-0.769) | 0.938 (0.935-0.941) | 0.694 (0.689-0.699) |
| 35 | 0.753 (0.749-0.757) | 0.940 (0.936-0.943) | 0.679 (0.674-0.684) |
| 40 | 0.736 (0.732-0.740) | 0.933 (0.930-0.936) | 0.660 (0.655-0.665) |

Table 29: Generative performance metrics as a function of the number of codes generated. The BERTScore F1 metric is a measure of cosine similarity between the uncontextualized generated and true **place-of-service** sequences. Error bars indicate bootstrapped 95% confidence intervals.

| Number of Codes | BERTScore F1 | BERTScore Precision | BERTScore Recall |
|---|---|---|---|
| 5 | 0.969 (0.968-0.970) | 0.985 (0.984-0.986) | 0.956 (0.955-0.958) |
| 10 | 0.968 (0.968-0.969) | 0.992 (0.991-0.993) | 0.948 (0.947-0.950) |
| 15 | 0.967 (0.966-0.968) | 0.995 (0.995-0.996) | 0.943 (0.942-0.944) |
| 20 | 0.966 (0.965-0.967) | 0.997 (0.996-0.997) | 0.940 (0.938-0.941) |
| 25 | 0.965 (0.964-0.965) | 0.997 (0.997-0.998) | 0.936 (0.935-0.938) |
| 30 | 0.963 (0.962-0.964) | 0.998 (0.998-0.998) | 0.932 (0.931-0.934) |
| 35 | 0.963 (0.962-0.963) | 0.999 (0.998-0.999) | 0.931 (0.930-0.932) |
| 40 | 0.961 (0.961-0.962) | 0.999 (0.998-0.999) | 0.929 (0.928-0.930) |