# OpenReview forum: "CHIRon: A Generative Foundation Model for Structured Sequential Medical Data"
_ICLR.cc/2025/Conference — ICLR 2025 Conference Withdrawn Submission_

### Official Review · Reviewer_rwhh · 2024-10-27

**Soundness:** 2
**Presentation:** 2
**Contribution:** 2
**Rating:** 3
**Confidence:** 3

**Summary:**

The paper presents CHIRon, a generative foundation model (FM) designed specifically for structured sequential medical data. As a decoder-only model, CHIRon is pre-trained with causal masking to enable generative tasks and uses diverse medical data types, including diagnoses, procedures, medications, lab values, demographics, and places of service. CHIRon introduces other auxiliary tasks, such as predicting the place of service and age at encounter, which improve the model's data generation quality and downstream performance. The authors explore multiple approaches to embedding continuous lab values, providing insights with potential applications for other FMs handling continuous data.

**Strengths:**

Data Scale: The model is trained on a large-scale medical dataset, allowing it to capture complex patterns within structured sequential data effectively.

Downstream Task Performance: CHIRon demonstrates promising results on downstream prediction tasks, showing improvements over the baseline models tested. This highlights its potential utility in various clinical applications.

Generative Capability for Medical Codes: The model is one of the first to tackle generative tasks with medical codes, effectively expanding the scope of clinical foundation models to generate and evaluate structured medical data sequences.

**Weaknesses:**

Lack of Novelty: The approach lacks clear originality, as autoregressive models for medical data, especially those incorporating procedure codes, medications, lab results, and demographics, are well-explored. Several models, such as CLMBR, MOTOR, EventStreamGPT, and ETHOS, also use decoder-only architectures with causal or autoregressive objectives. The one unique aspect in CHIRon is the addition of pretraining tasks (predicting place of service and age at encounter), but even these are present as direct features in models like CLMBR.

Limited Baseline Comparisons: The study does not include critical autoregressive baselines, such as CLMBR, which shares similarities in design as a decoder-only autoregressive model. This makes it challenging to assess the comparative performance of CHIRon accurately.

Questionable Utility of Additional Pretraining Tasks: While the model adds a pretraining task for predicting the place of service, the benefit of this task remains unclear, as embeddings for the place of service are already included. The redundancy and utility of this pretraining task are not well-demonstrated, especially given that other models like CLMBR and EventStreamGPT include similar objectives as part of their standard autoregressive prediction tasks.

Evaluation Limitations with Standard NLP Metrics: Using standard NLP metrics may not accurately capture the performance of medical code generation, as minor variations in medical codes can significantly change diagnostic meaning. Including qualitative results, such as expert-verified generated samples and gold-standard examples, would improve evaluation quality. Additionally, predictive metrics over long-term simulated trajectories (e.g., AUROC for one-year risk prediction) would better capture the model’s generative capabilities and clinical relevance. A generative model could be used to answer long term predictive problems by simulating a year of future data and seeing if a code appears in those simulated histories. One can run multiple trajectories to get a probability.

Suboptimal Empirical Results for Novel Contributions: The causal visit-based masking approach, claimed as a novel contribution, shows suboptimal performance in comparison to simple next-code prediction, as shown in Figure 4. This weakens the empirical support for the proposed masking method.

Potential Bias in Index Time Selection: It is not clear if the same index time selection procedure is used for both cases and controls, which can induce bias, as models might learn artifacts from the index time selection process rather than actual clinical features. Resources like (https://ohdsi.github.io/TheBookOfOhdsi/PatientLevelPrediction.html) provide guidance on ensuring consistent index time procedures for cases and controls. It would be beneficial for the authors to clarify the index time selection process for controls and address any potential biases introduced by differing procedures.

Lack of Transparency and Reproducibility: The model is trained and evaluated on a private dataset, and the code is not provided. Given the reliance on proprietary data, releasing the codebase would be essential for validation and further use of the model in research.

**Questions:**

End-to-End Fine-Tuning vs. Linear Probing: Why did the authors choose to fine-tune the entire FM for downstream tasks instead of using a linear head or conducting linear probing, which is more typical for foundation model evaluation? Could the authors provide results using linear probing to help isolate the quality of the learned representations?

Control Group Index Time Selection: The paper does not specify how index times are chosen for control patients. Could the authors clarify the method for selecting index times for controls? If this differs from the case group, how do the authors address potential bias introduced by this difference?

Evaluation Set for Generation Capabilities: For the evaluation of CHIRon’s generative capabilities, are the patients used for testing taken from a completely held-out set? Ensuring this separation would reinforce confidence in the model’s generative performance.

Place of Service Embedding and Pretraining Task: The model includes both an embedding for the place of service and a separate pretraining task to predict it. Given that place of service information is already embedded, what is the added benefit of predicting it as a pretraining task? A clarification here would help clarify the design choice and whether it provides an empirical advantage.

Qualitative and Expert Evaluation of Generated Medical Codes: Standard NLP metrics may not fully capture the accuracy of generated medical codes, as slight variations in codes can significantly alter diagnostic meaning. Could the authors provide qualitative examples, ideally evaluated by medical experts, comparing generated samples to the gold-standard sequences? This addition would strengthen the validity of the generative evaluations.

Generative Evaluation for Predictive Accuracy: It would be insightful to assess CHIRon’s utility for disease prediction by using its generative model to predict long-term risk. One way to do this could be by simulating patient trajectories over a year to estimate the likelihood of a disease code appearing in future sequences, yielding an AUROC for prediction accuracy. This would offer a robust, clinically relevant metric for evaluating the model's generative quality.

Supplemental Section and Code Availability: Given the model’s reliance on proprietary data, it would be highly beneficial if the authors could share the supplemental section and code for review, aiding transparency and reproducibility.

---

### Official Review · Reviewer_iRqB · 2024-10-29

**Soundness:** 2
**Presentation:** 2
**Contribution:** 2
**Rating:** 3
**Confidence:** 4

**Summary:**

The paper proposes CHIRon, a foundation model designed specifically for sequential medical data. CHIRon uses a generative approach with causal masking to handle diverse medical data types (e.g., diagnoses, lab results, demographics) and introduces a new pre-training objective for predicting various medical aspects. This model aims to generate realistic medical data sequences and improve performance on disease classification tasks.

**Strengths:**

1. Effective handling of heterogeneous EHR data types, particularly through lab test tokenization strategies.
2. Promising results on the large scale medical dataset

**Weaknesses:**

1. The paper has limited technology contributions to the computer science audience. The main contribution is about the numerical lab results tokenization. However, as a foundation model work, the generalizability of the method is unknown on other datasets, making it difficult to evaulate the methodology novelty
2. The main limitation lies in the generalizabiility of the model. The baseline models and evaluation tasks are limited compared to recent medical (EHR) foundation model works [1,2]. They are evaluated using tens of tasks or few-shot cross-dataset tasks. While the authors argue the reason not using MIMIC datasets, only predicting on five diseases is not convincing to show the generalizability of the model, which is crucial for foundation model works. Also the author fail to clarify why these five disesaes are selected.
3. The baseline models are also limited. The experiments should including recent foundation model works, and as well as traditional models such as LSTM and GRU to outline why using GPT-based architecture is preferred.
[1] Steinberg, E., Fries, J. A., Xu, Y., & Shah, N. MOTOR: A Time-to-Event Foundation Model For Structured Medical Records. In The Twelfth International Conference on Learning Representations.
[2] Wornow, M., Thapa, R., Steinberg, E., Fries, J., & Shah, N. (2023). Ehrshot: An ehr benchmark for few-shot evaluation of foundation models. Advances in Neural Information Processing Systems, 36, 67125-67137.

**Questions:**

see weakness

---

### Official Review · Reviewer_wags · 2024-11-03

**Soundness:** 3
**Presentation:** 3
**Contribution:** 2
**Rating:** 5
**Confidence:** 4

**Summary:**

Authors present CHIRon, a pretrained generative model for structured medical codes based on GPT2-like structure. Compared to previous works, CHIRon improved in several aspects. 1, data preprocessing (lab results are tokenized as well, embeddings for place of visit) 2, new pretraining tasks predicting place-of-service and age-at-encounter information. Authors demonstrate CHIRon’s generative capabilities for creating realistic patient records and downstream task performance.

**Strengths:**

1. Paper is well-written and motivated,
2. Structured EHRs is complicated for LLMs. Everything are explained well such as embeddings, tokenziation, pretraining-objective
3. Several experimental results support authors's claim, CHIRon achieved better generation capability as well as downstream performance.

**Weaknesses:**

1. Limited Technical Innovation: The architecture largely leverages existing approaches, using GPT-2 as its foundation
The tokenization strategy is based on Med-BERT because Med-BERT excludes lab results,The approach to incorporate lab results seems to be adapted from Golkar et al.'s work.
2. Potentially Misleading Claims: The abstract's characterization of a "new pre-training objective function" needs clarification
What's presented as a novel objective function is essentially still a next-code prediction task?
The main difference lies in preprocessing modifications rather than fundamental changes to the objective function itself.
3. Incomplete Evaluation: EHRSHOT is a closely related work and necessary to compare. More importantly, since authors consider CHIRon a foundation model,  it's necessary to shows its adaptation capability. Lack of transfer learning experiments in the paper.
4. Reproducibility question: Absence of publicly available code

**Questions:**

1. Comparison with TransformerEHR: As I know, there is an issue with its GitHub repository (missing essential implementation file)
How did authors implement the comparison?

---

### Official Review · Reviewer_hWZn · 2024-11-05

**Soundness:** 2
**Presentation:** 2
**Contribution:** 1
**Rating:** 3
**Confidence:** 4

**Summary:**

This paper evaluates methods for modeling Electronic Health Record (EHR) time series data, focusing specifically on comparing three tokenization approaches and three autoregressive pretraining tasks in the transfer learning setting. The authors do a limited comparison of the generated data quality for two of the autoregressive pretraining methods as well and only one of the tokenization methods.

**Strengths:**

1. Tokenization Comparison:
The authors compare three tokenization strategies in the zero-shot setting
    1. decile-binned tokenization is performed where lab numeric values are mapped to deciles and collapsed into the token vocabulary.
    2. They try decile-binned tokens where they multiply by a normalized continuous value of the lab
    3. Sum two embeddings where the first is for the medical code and the second is the decile bin

They find that tokenization method 1 consistently works best for finetuning and 3 is the worst. An interesting and practical result

It is clear they do some additional percentile binning, but it is not communicated well. I think a more systematic discussion of binning strategies (and presentation of the results) should be provided.


2. Pretraining Task Comparison
The authors compare three pretraining strategies vanilla autoregressive pretraining (the CHIRON model) with two additional modifications
    1. Vanilla autoregressive pretraining
    2. autoregressive pretraining with *added task heads* for predicting place of service and age
    3. autoregressive pretraining with *added task heads* as in (2) and with *Visit-Based Causal Masking*. So instead of typical causal masking, generating codes for a visit can only attend to the codes from past visits.


They find that that vanilla autoregressive pretraining works best in the finetuning setting. In the generative evaluation, they see that the added task heads improve generation performance. Why is Visit-based causal masking not evaluated in the generative setting?

3. Evaluation

The authors evaluate in two settings
a. Finetuning Evaluation:
They postpend a CLS token and use that representation to predict if a patient will have specific future diseases.


b. Generated Data Evaluation:
Bertscore (cosine similarity of generated tokens) and Rouge-1 (Overlap of unigrams). These metrics are not the most clinically significant metrics, but are a first step for assessing generated data quality in terms of similarity to the real sequence.

**Weaknesses:**

1. Missing Zero-shot classification Evaluation

Why did you decide to not investigate zero-shot binary classification performance in this work? This is one of the major applications in the field?

Several key related works provide frameworks or results evaluating autoregressive generative models on clinical data, and these works diminish the novelty of this paper:
1. ESGPT [1] is an autoregressively pretrained decoder-only transformer model that in the paper integrates age in tokens, and through it's user api can support user defined time specific features such as the place of service. The API allows zero-shot binary classification as well, something never touched upon in this work.
2. Ethos [2] is another model that performs decile binning and evaluates binary classification performance in the zero-shot setting. They actually run this on MIMICIV. It leaves out comparisons to baselines like XGBoost baselines, or supervised models though which you could improve upon.
3. Foresight [3] is another one of these autoregressive pretrained EHR data models you should cite.


2. Incomplete Analysis
- Generated Data Evaluation--tokenization method analysis: Why is there no comparison of the tokenization methods in the generated data evaluation setting?
- Custom percentile binning is not systematically assessed: I expected there to be a systematic assessment of well defined binning strategies for numeric values and specific takeaways on this, but there it no clear analysis/takeaway. Additionally I expected a comparison of no binning and encoding using only the continuous value of the lab, but also didn't find this.
- Sorting data - This paper seems to position itself as providing controlled experiments between tokenization methods and autoregressive pretraining tasks, developing an improved method from past works. One major limitation is that you always use random ordering of codes within visits. The affect of sorting codes within visits will have a significant affect on modeling. Next code prediction within a visit is significantly easier given a consistent ordering. This is precisely what ETHOS [1] does.

3. MIMICIV Experiments Should be Run

The authors do not run MIMIC IV experiments stating
> While we agree MIMIC-IV is a well known public resource for high quality patient data from ICU stays, both CHIRon and related foundational models focus on more general longitudinal healthcare data,not specific to ICU stays.

In fact MIMIC-IV v3.0 includes

364,627 patients, **546,028 inpatient admissions**, and 94,458 icustays. MIMIC-IV is generally a gold standard dataset for communicated ML results on in the ML literature, so the reasoning for not including it here is lackluster. The authors do use a much larger private dataset on the order of 25 million patients and do significant processing of this dataset, maybe a communication (in the limitations section) of the limitations and requirements of **the extensive data preprocessing for your model and evaluation** and how they make running experiments on MIMIC-IV out of scope for this paper is necessary.

4. Visit-based causal masking performing worse in the finetuning explanation is not a novel or interesting result.

Having access to previously generated tokens is not data leakage. An autoregressive generative model always is conditioned on past generated codes, so Visit-based causal modeling (where you mask out previous generated codes in the same visit) is obviously going to hurt model performance because model has to unconditionally generate codes without knowing which ones it already generated for the visit.

5. Insufficient Evaluation metrics for EHR data

The evaluation metrics used only assess similar marginal distributions of codes, which implies that ROUGE-1 is really not a clinically appropriate metric for assessing EHR generated data quality since it ignores sequential order. While BERTScore does implicitly capture sequential ordering in the contextual embeddings there should be additional metrics that explicitly evaluate sequential order in the medical context, such as metrics that consider the temporal relationships between generated medical events in different visits reflecting the real ordering across visits.



## Presentation issues

Please make the legend in Figure 3 clearer, and make concise names for the tokenization methods, that you can use in the legend. You should clarify what the ptile binning strategy is. I had to check the appendix and the intro to understand that you did some custom percentile binning, which is great, but should be clearer in the main text and you should give a systematic discussion of that these percentil binning strategies are and why they were chosen (per weakness 5).


[1] McDermott, Matthew, et al. "Event Stream GPT: a data pre-processing and modeling library for generative, pre-trained transformers over continuous-time sequences of complex events." Advances in Neural Information Processing Systems 36 (2023): 24322-24334.

[2] Renc, Pawel, et al. "Zero shot health trajectory prediction using transformer." NPJ Digital Medicine 7.1 (2024): 256.


[3] Kraljevic, Zeljko, et al. "Foresight—a generative pretrained transformer for modelling of patient timelines using electronic health records: a retrospective modelling study." The Lancet Digital Health 6.4 (2024): e281-e290.

**Questions:**

I will use this section to briefly summarize the top actionable things in my opinion.

1. Why do you not perform a zero-shot generation analysis where you compare binary classification performance for your finetuning tasks? This is the main use case for an autoregresive EHR foundation model right, to generate trajectories that actually reflect the real patient's disease progression.
2. Finish your analysis -- compare all tokenization methods and pretraining task methods in the generation setting
3. Add a sequential ordering metric (BertScore and Rough-1 do not explicitly assess sequential ordering accross sequences)
5. Why is there not a clearer and more systematic analysis for the different percentile binnings? I see in the appendix you have done this, but it just isn't presented clearly in figure 3, and is not a systemmatic analysis of different clearly communicated numeric value binning methods. You also are missing a strategy where you don't bin at all and just use an embedding of the raw continuous value. A demonstration of how binning (as opposed to a continuous embedding strategy) would be interesting.
6. Why even include the Visit-based masking baseline? It makes so much more sense to evaluate other potential auxilliary tasks than to do this experiment with an obvious result of hurting performance. And also to compare sorting codes within a visit vs random sorting. I know often we want to highlight all results and work in a paper (and it is evident you all did a lot of hard work), but this masking baseline really doesn't add anything to the paper as the negative result is obvious.

---

### Note · Authors · 2024-11-25

I have read and agree with the venue's withdrawal policy on behalf of myself and my co-authors.